# Variational Weighting for Kernel Density Ratios

**Sangwoong Yoon**
Korea Institute for Advanced Study
swyoon@kias.re.kr

**Frank C. Park**
Seoul National University / Saige Research
fcp@snu.ac.kr

**Gunsu Yun**
POSTECH
gunsu@postech.ac.kr

**Iljung Kim**
Hanyang University
iljung0810@hanyang.ac.kr

**Yung-Kyun Noh**
Hanyang University / Korea Institute for Advanced Study
nohyung@hanyang.ac.kr

## Abstract

Kernel density estimation (KDE) is integral to a range of generative and discriminative tasks in machine learning. Drawing upon tools from the multidimensional calculus of variations, we derive an optimal weight function that reduces bias in standard kernel density estimates for density ratios, leading to improved estimates of prediction posteriors and information-theoretic measures. In the process, we shed light on some fundamental aspects of density estimation, particularly from the perspective of algorithms that employ KDEs as their main building blocks.

## 1 Introduction

One fundamental component for building many applications in machine learning is a correctly estimated density for prediction and estimation tasks, with examples ranging from classification [1, 2], anomaly detection [3], and clustering [4] to the generalization of value functions [5], policy evaluation [6], and estimation of various information-theoretic measures [7, 8, 9]. Nonparametric density estimators, such as the nearest neighbor density estimator or kernel density estimators (KDEs), have been used as substitutes for the probability density component within the equation of the posterior probability, or the density-ratio equation, with theoretical guarantees derived in part from the properties of the density estimators used [10, 11].

Given a specific task which uses the ratio between two densities, $p_1(\mathbf{x})$ and $p_2(\mathbf{x})$ at a point $\mathbf{x} \in \mathbb{R}^D$, we consider the ratio handled by the ratio of their corresponding two KDEs, $\widehat{p}_1(\mathbf{x})$ and $\widehat{p}_2(\mathbf{x})$:

$$\frac{\widehat{p}_1(\mathbf{x})}{\widehat{p}_2(\mathbf{x})} \xrightarrow[\text{Estimate}]{} \frac{p_1(\mathbf{x})}{p_2(\mathbf{x})}. \tag{1}$$

Each estimator is a KDE which counts the effective number of data within a small neighborhood of $\mathbf{x}$ by averaging the kernels. The biases produced by the KDEs in the nominator and denominator [12, Theorem 6.28] are combined to produce a single bias of the ratio, as demonstrated in Fig. 1. For example, the ratio $\frac{p_1(\mathbf{x})}{p_2(\mathbf{x})}$ at $\mathbf{x}_0$ in Fig. 1(a) is clearly expected to be underestimated because of the dual effects in Fig. 1(b): the *under*estimation of the nominator $p_1(\mathbf{x}_0)$ and the *over*estimation of the denominator $p_2(\mathbf{x}_0)$. The underestimation is attributed to the concavity of $p_1(\mathbf{x})$ around $\mathbf{x}_0$ which leads to a reduced number of data being generated compared to a uniform density. The underestimation of $p_2(\mathbf{x})$ can be explained similarly. The second derivative—Laplacian—that creates

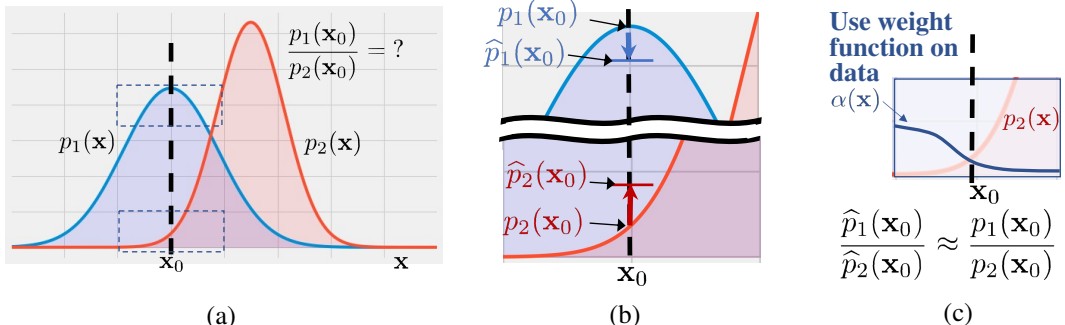

(a)  (b)  (c)

Figure 1: Estimation of the density ratio $\frac{p_1(\mathbf{x}_0)}{p_2(\mathbf{x}_0)}$ and bias correction using the weight function $\alpha(\mathbf{x})$. (a) Two density functions, $p_1(\mathbf{x})$ and $p_2(\mathbf{x})$, and the point of interest $\mathbf{x}_0$ for ratio estimation. Two regions delineated by dashed lines are magnified in (b). (b) The concavity and convexity of the density functions around $\mathbf{x}_0$ and their KDEs. Concave density $p_1(\mathbf{x})$ generates less data than the uniform density of $p_1(\mathbf{x}_0)$ around $\mathbf{x}_0$ resulting in an underestimation. For a similar reason, convex density $p_2(\mathbf{x})$ results in an overestimation. The two biases are combined into an underestimation of the ratio. (c) KDE augmented with a nonsymmetric weight function $\alpha(\mathbf{x})$ can alleviate this bias by transforming the bias of $\widehat{p}_2(\mathbf{x})$ to an appropriate underestimation from an overestimation.

the concavity or convexity of the underlying density is a dominant factor that causes the bias in this example.

The Laplacian of density has been used to produce equations in various bias reduction methods, such as bias correction [13, 14, 15] and smoothing of the data space [16, 17]. However, the example in Fig. 1 motivates a novel, position-dependent weight function $\alpha(\mathbf{x})$ to be multiplied with kernels in order to alleviate the bias. For example, to alleviate the overestimation of $p_2(\mathbf{x}_0)$, we can consider the $\alpha(\mathbf{x})$ shown in Fig. 1(c) that assigns more weight on the kernels associated with data located to the left of $\mathbf{x}_0$, which is a low-density region. When the weighted kernels are averaged, the overestimation of $\widehat{p}_2(\mathbf{x}_0)$ can be mitigated or potentially even underestimated. Meanwhile, the bias of $\widehat{p}_1(\mathbf{x}_0)$ remains unchanged after applying the weights since $p_1(\mathbf{x})$ is symmetric around $\mathbf{x}_0$. This allows the reversed underestimation of $p_2(\mathbf{x}_0)$ from the initial overestimation to effectively offset or counterbalance the underestimation of $p_1(\mathbf{x}_0)$ within the ratio.

We derive the $\alpha(\mathbf{x})$ function that performs this alleviation over the entire data space. The appropriate information for $\alpha(\mathbf{x})$ comes from the geometry of the underlying densities. The aforementioned principle of bias correction leads to novel, model-based and model-free approaches. Based on the assumption of the underlying densities, we learn the parameters for the densities' first and second derivatives and then variationally adjust $\alpha(\mathbf{x})$ for the estimator to create the variationally weighted KDE (VWKDE). We note that the model for those densities and their derivatives need not be exact because the goal is not to achieve precise density estimation but rather to accurately capture the well-behaved $\alpha(\mathbf{x})$ for the KDE ratios.

Applications include classification with posterior information and information-theoretic measure estimates using density-ratio estimation. Calibration of posteriors [18] has been of interest to many researchers, in part, to provide a ratio of correctness of the prediction. Plug-in estimators of information-theoretic measures, such as the Kullback-Leibler (K-L) divergence, can also be advantageous. For K-L divergence estimation, similar previous formulations for the variational approach have included optimizing a functional bound with respect to the function constrained within the reproducing kernel Hilber space (RKHS) [19, 20, 21]. These and other methods that use weighted kernels (e.g., [22, 23, 24]) take advantage of the flexibility offered by universal approximator functions in the form of linear combinations of kernels. These methods, however, do not adequately explain why the weight optimization leads to an improved performance. Based on a derivation of how bias is produced, we provide an explicit modification of weight for standard kernel density estimates, with details of how the estimation is improved.

The remainder of the paper is organized as follows. In Section 2, we introduce the variational formulation for the posterior estimator and explain how to minimize the bias. Section 3 shows how a weight function can be derived using the calculus of variations, which is then extended to general density-ratio and K-L divergence estimation in Section 4. Experimental results are presented in Section 5. Finally, we conclude with discussion in Section 6.

## 2 Variationally Weighted KDE for Ratio Estimation

KDE $\widehat{p}(\mathbf{x}) = \frac{1}{N}\sum_{j=1}^{N} k_h(\mathbf{x}, \mathbf{x}_j)$ is conventionally the average of kernels. The average roughly represents the count of data within a small region around $\mathbf{x}$, the size of which is determined by a bandwidth parameter $h$. The amount of convexity and concavity inside the region determines the bias of estimation, as depicted in Fig. 1(a),(b).

### 2.1 Plug-in estimator of posterior with weight

We consider a weighted KDE as a plug-in component adjusted for reliable ratio estimation using a positive and twice differentiable weight function: $\alpha(\mathbf{x}) \in \mathcal{A}$ with $\mathcal{A} = \{\alpha : \mathbb{R}^D \to \mathbb{R}^+ \mid \alpha \in C^2(\mathbb{R}^D)\}$. For two given sets of i.i.d. samples, $\mathcal{D}_1 = \{\mathbf{x}_i\}_{i=1}^{N_1} \sim p_1(\mathbf{x})$ for class $y = 1$ and $\mathcal{D}_2 = \{\mathbf{x}_i\}_{i=N_1+1}^{N_1+N_2} \sim p_2(\mathbf{x})$ for class $y = 2$, we use the following weighted KDE formulation:

$$\widehat{p_1}(\mathbf{x}) = \frac{1}{N_1}\sum_{j=1}^{N_1} \alpha(\mathbf{x}_j)k_h(\mathbf{x}, \mathbf{x}_j), \quad \widehat{p_2}(\mathbf{x}) = \frac{1}{N_2}\sum_{j=N_1+1}^{N_1+N_2} \alpha(\mathbf{x}_j)k_h(\mathbf{x}, \mathbf{x}_j). \tag{2}$$

Here, the two estimators use a single $\alpha(\mathbf{x})$. The kernel function $k_h(\mathbf{x}, \mathbf{x}')$ is a positive, symmetric, normalized, isotropic, and translation invariant function with bandwidth $h$. The weight function $\alpha(\mathbf{x})$ informs the region that should be emphasized, and a constant function $\alpha(\mathbf{x}) = c$ reproduces the ratios from the conventional KDE. We let their plug-in posterior estimator be $f(\mathbf{x})$, and the function can be calculated using

$$f(\mathbf{x}) = \widehat{P}(y = 1|\mathbf{x}) = \frac{\widehat{p_1}(\mathbf{x})}{\widehat{p_1}(\mathbf{x}) + \gamma\widehat{p_2}(\mathbf{x})}. \tag{3}$$

with a constant $\gamma \in \mathbb{R}$ determined by the class-priors.

### 2.2 Bias of the posterior estimator

We are interested in reducing the expectation of the bias square:

$$\mathbb{E}[\text{Bias}(\mathbf{x})^2] = \int \left(f(\mathbf{x}) - \mathbb{E}_{\mathcal{D}_1, \mathcal{D}_2}[f(\mathbf{x})]\right)^2 p(\mathbf{x})d\mathbf{x}. \tag{4}$$

The problem of finding the optimal weight function can be reformulated as the following equation in Proposition 1.

**Proposition 1.** *With small $h$, the expectation of the bias square in Eq. (4) is minimized by any $\alpha(\mathbf{x})$ that eliminates the following function*

$$B_{\alpha;p_1,p_2}(\mathbf{x}) = (\nabla \log \alpha|_{\mathbf{x}})^\top \mathbf{h}(\mathbf{x}) + g(\mathbf{x}), \tag{5}$$

*at every point $\mathbf{x}$. Here, $\mathbf{h}(\mathbf{x}) = \left(\frac{\nabla p_1}{p_1} - \frac{\nabla p_2}{p_2}\right)$ and $g(\mathbf{x}) = \frac{1}{2}\left(\frac{\nabla^2 p_1}{p_1} - \frac{\nabla^2 p_2}{p_2}\right)$ with gradient and Laplacian operators, $\nabla$ and $\nabla^2$, respectively. All derivatives are with respect to $\mathbf{x}$.* ∎

The derivation of Eq. (5) begins with the expectation of the weighted KDE:

$$\mathbb{E}_{\mathcal{D}_1}[\widehat{p_1}(\mathbf{x})] = \mathbb{E}_{\mathbf{x}'\sim p_1(\mathbf{x})}[\alpha(\mathbf{x}')k_h(\mathbf{x}, \mathbf{x}')] = \int \alpha(\mathbf{x}')p_1(\mathbf{x}')k_h(\mathbf{x}, \mathbf{x}')\mathrm{d}\mathbf{x}' \tag{6}$$

$$= \alpha(\mathbf{x})p_1(\mathbf{x}) + \frac{h^2}{2}\nabla^2[\alpha(\mathbf{x})p_1(\mathbf{x})] + O(h^3). \tag{7}$$

Along with the similar expansion for $\mathbb{E}_{\mathcal{D}_2}[\widehat{p_2}(\mathbf{x})]$, the following plug-in posterior can be perturbed with small $h$:

$$\mathbb{E}_{\mathcal{D}_1, \mathcal{D}_2}[f(\mathbf{x})] \to \frac{\mathbb{E}_{\mathcal{D}_1}[\widehat{p_1}(\mathbf{x})]}{\mathbb{E}_{\mathcal{D}_1}[\widehat{p_1}(\mathbf{x})] + \gamma\mathbb{E}_{\mathcal{D}_2}[\widehat{p_2}(\mathbf{x})]} \tag{8}$$

$$= f(\mathbf{x}) + \frac{h^2}{2}\frac{\gamma p_1(\mathbf{x})p_2(\mathbf{x})}{(p_1(\mathbf{x}) + \gamma p_2(\mathbf{x}))^2}\left(\frac{\nabla^2[\alpha(\mathbf{x})p_1(\mathbf{x})]}{\alpha(\mathbf{x})p_1(\mathbf{x})} - \frac{\nabla^2[\alpha(\mathbf{x})p_2(\mathbf{x})]}{\alpha(\mathbf{x})p_2(\mathbf{x})}\right) + \mathcal{O}(h^3)$$

$$= f(\mathbf{x}) + \frac{h^2}{2}P(y = 1|\mathbf{x})P(y = 2|\mathbf{x})B_{\alpha;p_1,p_2}(\mathbf{x}) + \mathcal{O}(h^3), \tag{9}$$

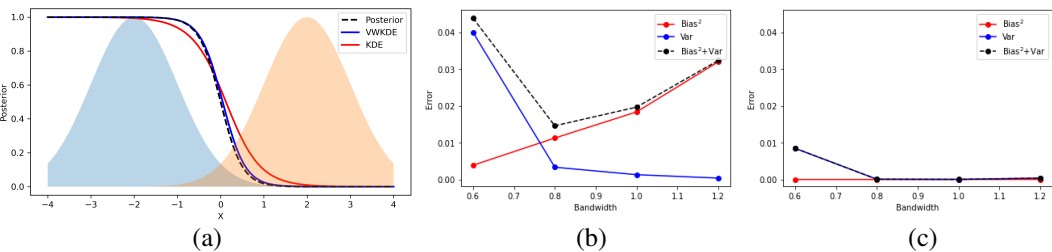

Figure 2: Posterior estimates with KDEs and VEKDEs for two 20-dimensional homoscedastic Gaussian densities. (a) Bias corrected by VWKDE. (b) Bias and variance of posterior estimates depending on the bandwidth for standard KDE and for (c) VWKDE.

with the substitution

$$B_{\alpha;p_1,p_2}(\mathbf{x}) \equiv \frac{\nabla^2[\alpha(\mathbf{x})p_1(\mathbf{x})]}{\alpha(\mathbf{x})p_1(\mathbf{x})} - \frac{\nabla^2[\alpha(\mathbf{x})p_2(\mathbf{x})]}{\alpha(\mathbf{x})p_2(\mathbf{x})}. \tag{10}$$

The point-wise leading-order bias can be written as

$$\text{Bias}(\mathbf{x}) = \frac{h^2}{2}P(y=1|\mathbf{x})P(y=2|\mathbf{x})B_{\alpha;p_1,p_2}(\mathbf{x}). \tag{11}$$

Here, the $B_{\alpha;p_1,p_2}(\mathbf{x})$ includes the second derivative of $\alpha(\mathbf{x})p_1(\mathbf{x})$ and $\alpha(\mathbf{x})p_2(\mathbf{x})$. Because two classes share the weight function $\alpha(\mathbf{x})$, Eq. (10) can be simplified into two terms without the second derivative of $\alpha(\mathbf{x})$ as

$$B_{\alpha;p_1,p_2}(\mathbf{x}) = \frac{\nabla^\top\alpha|_{\mathbf{x}}}{\alpha(\mathbf{x})}\left(\frac{\nabla p_1|_{\mathbf{x}}}{p_1(\mathbf{x})} - \frac{\nabla p_2|_{\mathbf{x}}}{p_2(\mathbf{x})}\right) + \frac{1}{2}\left(\frac{\nabla^2 p_1|_{\mathbf{x}}}{p_1(\mathbf{x})} - \frac{\nabla^2 p_2|_{\mathbf{x}}}{p_2(\mathbf{x})}\right), \tag{12}$$

which leads to Eq. (5) in the Proposition. The detailed derivation in this section can be found in Appendix A.

### 2.3 Plug-in estimator of K-L divergence with weight

In order to estimate $KL(p_1||p_2) = \mathbb{E}_{\mathbf{x}\sim p_1}\left[\log\frac{p_1(\mathbf{x})}{p_2(\mathbf{x})}\right]$, we consider the following plug-in estimator:

$$\widehat{KL}(p_1||p_2) = \frac{1}{N_1}\sum_{i=1}^{N_1}\log\frac{\widehat{p}_1(\mathbf{x}_i)}{\widehat{p}_2(\mathbf{x}_i)}, \tag{13}$$

using $\mathbf{x}_i$ in $\mathbf{x}_i \in \mathcal{D}_1$ for Monte Carlo averaging. When we calculate $\widehat{p}_1$ at $\mathbf{x}_i$, we exclude $\mathbf{x}_i$ from the KDE samples. We use $\widehat{p}_1(\mathbf{x}_i) = \frac{1}{N_1-1}\sum_{j=1}^{N_1}\alpha(\mathbf{x}_j)k_h(\mathbf{x}_i,\mathbf{x}_j)\mathbf{I}_{(i\neq j)}$ with the indicator function $\mathbf{I}_{(\mathcal{I})}$, which is 1 if $\mathcal{I}$ is true and 0 otherwise.

**Proposition 2.** *With small $h$, the expectation of the bias square is minimized by finding any $\alpha(\mathbf{x})$ that eliminates the same function as $B_{\alpha;p_1,p_2}(\mathbf{x})$ in Eq. (5) in Proposition 1 at each point $\mathbf{x}$.* ∎

In the task of estimating the K-L divergence, Proposition 2 claims that we obtain the equivalent $B_{\alpha;p_1,p_2}(\mathbf{x})$ to Eq. (5) during the derivation of bias. The pointwise bias in the K-L divergence estimator can be written as

$$\text{Bias}(\mathbf{x}) = \frac{h^2}{2}\frac{p_1(\mathbf{x})}{p_2(\mathbf{x})}B_{\alpha;p_1,p_2}(\mathbf{x}), \tag{14}$$

with $B_{\alpha;p_1,p_2}(\mathbf{x})$ equivalent to Eq. (5).

## 3 Variational Formulation and Implementation

Now we consider the $\alpha(\mathbf{x})$ that minimizes the mean square error for the estimation:

$$\min_{\alpha(\mathbf{x})\in\mathcal{A}}\int\left((\nabla\log\alpha|_{\mathbf{x}})^\top\mathbf{h}(\mathbf{x}) + g(\mathbf{x})\right)^2 r(\mathbf{x})d\mathbf{x}, \tag{15}$$

with $\mathbf{h}(\mathbf{x}) = \left( \frac{\nabla p_1}{p_1} - \frac{\nabla p_2}{p_2} \right)$, $g(\mathbf{x}) = \frac{1}{2} \left( \frac{\nabla^2 p_1}{p_1} - \frac{\nabla^2 p_2}{p_2} \right)$. The $r(\mathbf{x})$ function depends on the problem: $r(\mathbf{x}) = P(y=1|\mathbf{x})^2 P(y=2|\mathbf{x})^2 p(\mathbf{x})$ for posterior estimation and $r(\mathbf{x}) = \left( \frac{p_1(\mathbf{x})}{p_2(\mathbf{x})} \right)^2 p(\mathbf{x})$ for K-L divergence estimation, with the total density, $p(\mathbf{x}) = (p_1(\mathbf{x}) + \gamma p_2(\mathbf{x})) P(y=1)$. The calculus of variation for optimizing the functional in Eq. (15) provides an equation that the optimal $\alpha(\mathbf{x})$ should satisfy:

$$\nabla \cdot \left[ r((\nabla \log \alpha)^\top \mathbf{h} + g)\mathbf{h} \right] = 0. \tag{16}$$

The detailed derivation of this equation can be found in Appendix B.

## 3.1 Gaussian density and closed-form solution for $\alpha(\mathbf{x})$

A simple analytic solution for this optimal condition can be obtained for two homoscedastic Gaussian density functions. The density functions have two different means, $\mu_1 \in \mathbb{R}^D$ and $\mu_2 \in \mathbb{R}^D$, but share a single covariance matrix $\Sigma \in \mathbb{R}^{D \times D}$:

$$p_1(\mathbf{x}) = \mathcal{N}(\mathbf{x}; \mu_1, \Sigma), \quad p_2(\mathbf{x}) = \mathcal{N}(\mathbf{x}; \mu_2, \Sigma). \tag{17}$$

One solution for this homoscedastic setting can be obtained as

$$\alpha(\mathbf{x}) = \exp\left( -\frac{1}{2}(\mathbf{x} - \mu')^\top A (\mathbf{x} - \mu') \right), \tag{18}$$

with $\mu' = \frac{\mu_1 + \mu_2}{2}$ and $A = b \left( I - \frac{\Sigma^{-1}(\mu_1 - \mu_2)(\mu_1 - \mu_2)^\top \Sigma^{-1}}{\|\Sigma^{-1}(\mu_1 - \mu_2)\|^2} \right) - \Sigma^{-1}$ using an arbitrary constant $b$. Due to the choice of $b$, the solution is not unique. All the solutions produce a zero bias. Its detailed derivation can be found in the Appendix C. The reduction of the bias using Eq. (18) with estimated parameters is shown in Fig. 2.

## 3.2 Implementation

In this work, we propose a model-free method and a mode-based method. The model-free approach uses the information of $\widehat{\nabla} \log p_1(\mathbf{x})$ and $\widehat{\nabla} \log p_2(\mathbf{x})$ estimated by a score matching neural network [25]. We obtain the second derivative, $\widehat{\nabla}^2 \log p$, by the automatic differentiation of the neural network for the scores. We then obtain $\widehat{\frac{\nabla^2 p}{p}}$ using $\widehat{\frac{\nabla^2 p}{p}} = \widehat{\nabla}^2 \log p - \widehat{\nabla}^\top \log p \widehat{\nabla} \log p$. With the outputs of the neural networks for $\widehat{\nabla} \log p$ and $\widehat{\frac{\nabla^2 p}{p}}$, we train a new network for the function $\alpha(\mathbf{x}; \theta)$ with neural network parameters $\theta$.

On the other hand, the model-based approach uses a coarse Gaussian model for class-conditional densities. The Gaussian functions for each class have their estimated parameters $\widehat{\mu}_1, \widehat{\mu}_2 \in \mathbb{R}^D$ and $\widehat{\Sigma}_1, \widehat{\Sigma}_2 \in \mathbb{R}^{D \times D}$. We use the score information from these parameters: $\widehat{\nabla} \log p_1(\mathbf{x}) = \widehat{\Sigma}_1^{-1}(\mathbf{x} - \widehat{\mu}_1)$ and $\widehat{\nabla} \log p_2(\mathbf{x}) = \widehat{\Sigma}_2^{-1}(\mathbf{x} - \widehat{\mu}_2)$. In the model-based approach, we let the log of $\alpha(\mathbf{x})$ be a function within the RKHS with basis kernels $\kappa_\sigma(\cdot, \cdot)$ with kernel parameter $\sigma$. We let $\log \alpha(\mathbf{x}; \theta) = \sum_{i=1}^{N_1+N_2} \theta_i \kappa_\sigma(\mathbf{x}, \mathbf{x}_i)$ with parameters $\theta = \{\theta_1, \ldots, \theta_{N_1+N_2}\}$ for optimization.

The weight function $\alpha(\mathbf{x})$ is obtained by optimizing the following objective function with $N_1$ number of data from $p_1(\mathbf{x})$ and $N_2$ number of data from $p_2(\mathbf{x})$:

$$L(\theta) = \sum_{i=1}^{N_1+N_2} \frac{1}{2} \left( \nabla^\top \log \alpha(\mathbf{x}_i; \theta) \widehat{\mathbf{h}}(\mathbf{x}_i) \right)^2 + \nabla^\top \log \alpha(\mathbf{x}_i; \theta) \widehat{\mathbf{h}}(\mathbf{x}_i) \widehat{g}(\mathbf{x}_i), \tag{19}$$

with the substitutions $\widehat{\mathbf{h}}(\mathbf{x}) = \widehat{\nabla} \log p_1(\mathbf{x}) - \widehat{\nabla} \log p_2(\mathbf{x})$ and $\widehat{g}(\mathbf{x}) = \frac{1}{2} \left( \widehat{\frac{\nabla^2 p_1(\mathbf{x})}{p_1(\mathbf{x})}} - \widehat{\frac{\nabla^2 p_2(\mathbf{x})}{p_2(\mathbf{x})}} \right)$.

In the model-based method, an addition of $\ell_2$−regularizer, $\lambda \sum_{i=1}^{N_1+N_2} \theta_i^2$, with a small positive constant $\lambda$ makes the optimization (19) quadratic. When there are fewer than 3,000 samples, we use all of them as basis points. Otherwise, we randomly sample 3,000 points from $\{\mathbf{x}_i\}_{i=1}^{N_1+N_2}$.

A brief summary of the implementation process is shown in Algorithms 1 and 2.[1]

---

[1]Code is available at
https://github.com/swyoon/variationally-weighted-kernel-density-estimation

| **Algorithm 1** Model-free | **Algorithm 2** Model-based |
|---|---|
| **Input:** $\mathbf{x}$, $\{\mathbf{x}_i\}_{i=1}^{N_1} \sim p_1$, $\{\mathbf{x}_i\}_{i=N_1+1}^{N_1+N_2} \sim p_2$ | **Input:** $\mathbf{x}$, $\{\mathbf{x}_i\}_{i=1}^{N_1} \sim p_1$, $\{\mathbf{x}_i\}_{i=N_1+1}^{N_1+N_2} \sim p_2$ |
| **Output:** Ratio $\widehat{R}(\mathbf{x})$ $(=\widehat{p}_1/\widehat{p}_2(\mathbf{x}))$ | **Output:** Ratio $\widehat{R}(\mathbf{x})$ $(=\widehat{p}_1/\widehat{p}_2(\mathbf{x}))$ |
| **Procedure:** | **Procedure:** |
| 1. Estimate $\widehat{\nabla} \log p_1$ using $\{\mathbf{x}_i\}_{i=1}^{N_1} \sim p_1$. | 1. Estimate $\widehat{\mu}_1, \widehat{\Sigma}_1$ using $\{\mathbf{x}_i\}_{i=1}^{N_1} \sim p_1$. |
| 2. Estimate $\widehat{\nabla} \log p_2$ using $\{\mathbf{x}_i\}_{i=N_1+1}^{N_1+N_2} \sim p_2$. | 2. Estimate $\widehat{\mu}_2, \widehat{\Sigma}_2$ using $\{\mathbf{x}_i\}_{i=N_1+1}^{N_1+N_2} \sim p_2$. |
| 3. Obtain $\alpha(\mathbf{x})$ that minimizes Eq. (19) | 3. Use $\widehat{\nabla} \log p_c\|_{\mathbf{x}} = \widehat{\Sigma}_c^{-1}(\mathbf{x} - \widehat{\mu}_c)$ to obtain $\alpha(\mathbf{x})$ that minimizes Eq. (19) |
| 3. $\widehat{R}(\mathbf{x}) = \frac{\sum_{i=1}^{N_1} \alpha(\mathbf{x}_i) k_h(\mathbf{x},\mathbf{x}_i)}{\sum_{i=N_1+1}^{N_1+N_2} \alpha(\mathbf{x}_i) k_h(\mathbf{x},\mathbf{x}_i)}$ | 3. $\widehat{R}(\mathbf{x}) = \frac{\sum_{i=1}^{N_1} \alpha(\mathbf{x}_i) k_h(\mathbf{x},\mathbf{x}_i)}{\sum_{i=N_1+1}^{N_1+N_2} \alpha(\mathbf{x}_i) k_h(\mathbf{x},\mathbf{x}_i)}$ |
| **Return** $\widehat{R}(\mathbf{x})$ | **Return** $\widehat{R}(\mathbf{x})$ |

# 4 Interpretation of $\alpha(\mathbf{x})$ for Bias Reduction

The process of finding $\alpha(\mathbf{x})$ that minimizes the square of $B_{\alpha;p_1,p_2}(\mathbf{x})$ in Eq. (5) can be understood from various perspectives through reformulation.

## 4.1 Cancellation of the bias

The second term $\frac{1}{2}\left(\frac{\nabla^2 p_1}{p_1} - \frac{\nabla^2 p_2}{p_2}\right)$ in Eq. (5) repeatedly appears in the bias of nonparametric processes using discrete labels [26]. In our derivation, the term is achieved with a constant $\alpha(\mathbf{x})$ or with no weight function. The role of the weight function is to control the first term $\frac{\nabla^\top \alpha}{\alpha}\left(\frac{\nabla p_1}{p_1} - \frac{\nabla p_2}{p_2}\right)$ based on the gradient information in order to let the first term cancel the second.

## 4.2 Cancellation of flow in a mechanical system

The equation for each class can be compared with the mechanical convection-diffusion equation, $\frac{\partial u}{\partial t} = -\mathbf{v}^\top \nabla u + D'\nabla^2 u$, which is known as the equation for Brownian motion under gravity [27] or the advective diffusion equation of the incompressible fluid [28]. In the equation, $u$ is the mass of the fluid, $t$ is the time, $\mathbf{v}$ is the direction of convection, and $D'$ is the diffusion constant. The amount of mass change is the sum of the convective movement of mass along the direction opposite to $\nabla u$ and the diffusion from the neighborhood. We reorganize Eq. (5) into the following equation establishing the difference between the convection-diffusion equations of two fluids:

$$
\begin{aligned}
B_{\alpha;p_1,p_2}(\mathbf{x}) \;=\; & \left[\nabla^\top\left(\log \alpha + \frac{1}{2}\log p_1\right)\nabla \log p_1 + \frac{1}{2}\nabla^2 \log p_1\right] \\
& - \left[\nabla^\top\left(\log \alpha + \frac{1}{2}\log p_2\right)\nabla \log p_2 + \frac{1}{2}\nabla^2 \log p_2\right].
\end{aligned} \tag{20}
$$

According to the equation, we can consider the two different fluid mass functions, $u_1(\mathbf{x}) = \log p_1(\mathbf{x})$ and $u_2 = \log p_2(\mathbf{x})$, and the original convection movement along the directions $\mathbf{v}'_1 = -\frac{1}{2}\nabla \log p_1$ and $\mathbf{v}'_2 = -\frac{1}{2}\nabla \log p_2$. If we make an $\alpha(\mathbf{x})$ that modifies the convection directions $\mathbf{v}'_1$ and $\mathbf{v}'_2$ to $\mathbf{v}_1 = \mathbf{v}'_1 - \nabla\alpha$ and $\mathbf{v}_2 = \mathbf{v}'_2 - \nabla\alpha$, and a mass change in one fluid is compensated by the change of the other, in other words, if $\frac{\partial u_1}{\partial t} = \frac{\partial u_2}{\partial t}$, then the $\alpha(\mathbf{x})$ is the weight function that minimizes the leading term of the bias for ratio estimation.

## 4.3 Prototype modification in reproducing kernel Hilbert space (RKHS)

A positive definite kernel function has its associated RKHS. A classification using the ratio of KDEs corresponds to a prototype classification in RKHS that determines which of the two classes has a closer mean than the other [29, Section 1.2]. The application of a weight function corresponds to finding a different prototype from the mean [30]. The relationship between the new-found prototype and the KDEs has been previously discussed [31].

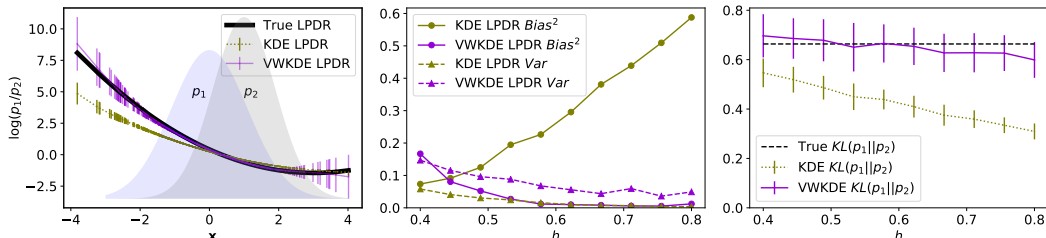

Figure 3: Estimation results of the LPDR $\log(p_1/p_2)$ and the K-L divergence. (a) Estimation of LPDR at each point. The estimation bias from the true LPDR is reduced by using VWKDE. (b) Squared bias and variance of the estimation with respect to the bandwidth $h$. Bias has been significantly reduced without increasing variance. (c) Mean and standard deviation of K-L divergence estimates with respect to the bandwidth $h$.

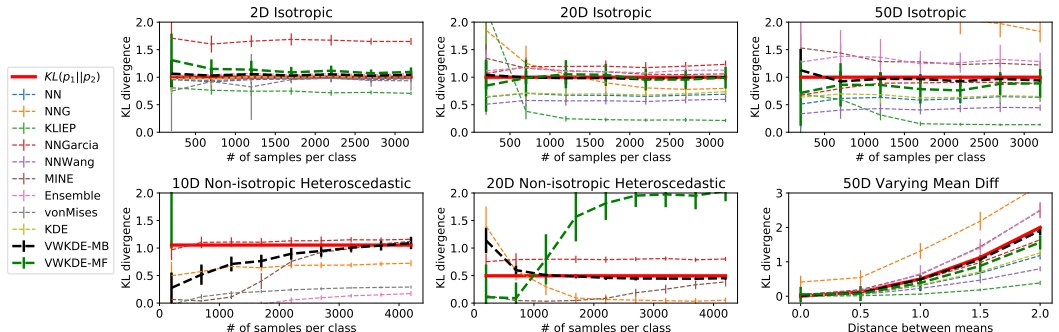

Figure 4: K-L divergence estimation results for synthetic distributions; (NN) Nearest-neighbor estimator; (NNG) NN estimator with metric learning [32]; (KLIEP) Direct importance estimation [21]; (NNGarcia) Risk-based $f$-divergence estimator [33]; (NNWang) Bias-reduced NN estimator [34]; (MINE) Mutual Information Neural Estimation [35]; (Ensemble) Weighted ensemble KDE estimator [36]; (vonMises) KDE estimator with von Mises expansion bias correction [37]; (KDE) KDE estimator; (VWKDE-MB, VWKDE-MF) Model-based and model-free approach of the proposed estimator in this paper.

## 5 Experiments

### 5.1 Estimation of log probability density ratio and K-L divergence in 1D

We first demonstrate in Fig. 3 how the use of VWKDE alters log probability density ratio (LPDR) and K-L divergence toward a better estimation. We use two 1-dimensional Gaussians, $p_1(x)$ and $p_2(x)$, with means 0 and 1 and variances $1.1^2$ and $0.9^2$, respectively. We draw 1,000 samples from each density and construct KDEs and model-based VWKDEs for both LPDR and K-L divergence. For LPDR evaluation, we draw a separate 1,000 samples from each density, and the average square of biases and the variances at those points are calculated and presented in Fig. 3(b). The K-L divergence estimation result is shown in Fig. 3(c), where the true K-L divergence can be calculated analytically as $KL(p_1||p_2) \approx 0.664$, and the estimated values are compared with this true K-L divergence.

KDE-based LPDR estimation exhibits a severe bias, but this is effectively reduced by using VWKDE as an alternative plug-in. Although VWKDE slightly increases the variance of estimation, the reduction of bias is substantial in comparison. Note that since the bias is small over a wide range of $h$, VWKDE yields a K-L divergence estimate which is relatively insensitive to the choice of $h$.

### 5.2 Synthetic distributions

We perform the VWKDE-based K-L divergence estimator along with other state-of-the-art estimators to estimate the K-L divergence $KL(p_1||p_2)$ between two synthetic Gaussian distributions $p_1$ and $p_2$ having $\mu_1$ and $\mu_2$ as their mean vectors and $\Sigma_1$ and $\Sigma_2$ as their covariance matrices, respectively.

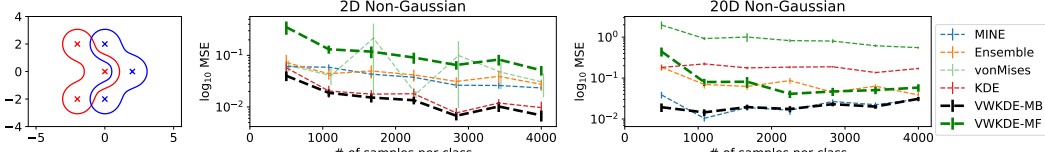

Figure 5: Estimation of K-L divergence between two non-Gaussian densities, $p_1(\mathbf{x})$ and $p_2(\mathbf{x})$. Each density is the Gaussian mixture of the three Gaussians, as shown in the 2-dimensional density contour in the figure on the left. They are the true densities but are very dissimilar to the single Gaussian model. The figure in the middle shows the estimation with 2-dimensional data, and the figure on the right shows the estimation with 20-dimensional data. With 20-dimensional data, the remaining 18 dimensionalities have the same mean isotropic Gaussians without correlation to the first two dimensionalities.

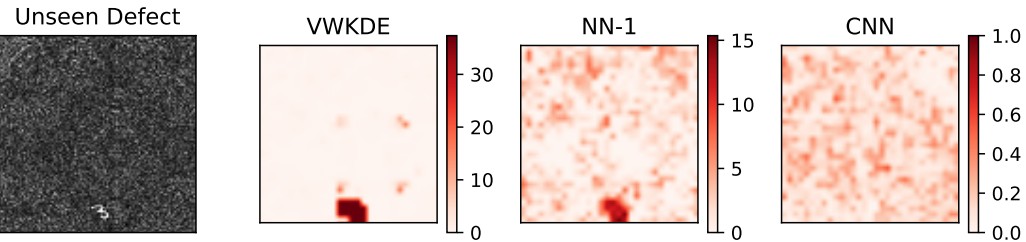

Figure 6: Detection of an artificially injected defect (MNIST digit "3"). The first panel shows the image with an injected defect. The remaining three panels show the detection scores of different methods.

We use three different settings for $p_1$ and $p_2$: Isotropic (**Iso**), Non-isotropic Heteroscedastic (**NH**), and Varying Mean Diff (**VMD**). In **Iso**, $\Sigma_1 = \Sigma_2 = I$ for an identity matrix $I$. $\mu_1 = \mathbf{0}$ for a zero vector $\mathbf{0}$. The first element of $\mu_2$ is $\sqrt{2}$, while the other elements are uniformly zero, resulting in $KL(p_1\|p_2) = 1$. In **NH**, $\mu_1 = \mu_2 = \mathbf{0}$, and $\Sigma_1 = \mathbf{I}$. $\Sigma_2$ is a matrix having a pair of off-diagonal element $(\Sigma_2)_{1,2} = (\Sigma_2)_{2,1} = 0.1$, and other off-diagonal elements are zero. The diagonal elements have a constant value $\omega$, which is determined to yield $KL(p_1\|p_2) \approx 1.0$ with $\omega = 0.750^2$ (10D) and $KL(p_1\|p_2) \approx 0.5$ with $\omega = 0.863^2$ (20D). In **VMD**, the first element of $\mu_2$ has various values between 0 and 2, while $\mu_1 = \mathbf{0}$ and the other elements in $\mu_2$ remain zero. $\Sigma_1 = \Sigma_2 = I$. In **Iso** and **NH**, we vary the sample size, and in **VMD**, we use 2,000 samples per distribution. We repeat each experiment 30 times and display the mean and the standard deviation in Figure 4.

In the upper left panel of Figure 4, we observe that almost all algorithms estimate the low-dimensional K-L divergence reliably, but the results deteriorate dramatically as shown in the upper middle and right panels with high-dimensionality. As shown in the lower left and lower middle panels, most of the baseline methods fail to produce the estimates near the true value when data are correlated. The model-based VWKDE-based estimator is the only estimator that recovers the true value in the 20D **NH** case. Figure 5 shows the K-L divergence estimation for non-Gaussian densities. In this example, the model for the score function in model-based VWKDE is different from the data-generating density, in which the estimator still shows very reliable estimates.

## 5.3 Unsupervised optical surface inspection

We apply the proposed K-L divergence estimation using VWKDE for the inspection of the surface integrity based on the optical images. Most of the previous works have formulated the inspection using the *supervised* setting [38, 39, 40, 41]; however, often the defect patterns are diverse, and training data do not include all possible defect patterns. In real applications, identification and localization of *unseen* defect patterns are important. In this example, we apply the model-based VWKDE.

**Detection of defective surface**   Following the representations of previous works on image classification [42, 43], we extract random small patches from each image $\mathbf{I}$ and assume that those patches are the independently generated data. We use the probability density $p_{\mathbf{I}}(\mathbf{x})$, for the patch $\mathbf{x} \in \mathbb{R}^D$ from $\mathbf{I}$. Given the $N$ number of normal surface images $\mathcal{D} = \{\mathbf{I}_i\}_{i=1}^N$ and a query image $\mathbf{I}^*$, we determine whether $\mathbf{I}^*$ is a defective surface according to the following decision function $f(\mathbf{I}^*)$ and a predefined

Table 1: Performances for defect surface detection (left) and defect localization (right). mAUC and mAP are averaged over six surface types of DAGM. The DAGM dataset is provided with labels, and only CNN used the labels for training. The unseen defect is the artificially injected MNIST digit "3."

| mAUC | DAGM Defect | Unseen Defect | mAP | DAGM Defect | Unseen Defect |
|------|-------------|---------------|-----|-------------|---------------|
| **VWKDE** | **0.785 ± 0.002** | **0.967 ± 0.003** | **VWKDE** | **0.369 ± 0.005** | **0.903 ± 0.007** |
| KDE | 0.734 ± 0.005 | 0.926 ± 0.003 | KDE | 0.294 ± 0.004 | 0.849 ± 0.006 |
| NN-1 | 0.628 ± 0.002 | 0.813 ± 0.001 | NN-1 | 0.095 ± 0.008 | 0.488 ± 0.002 |
| NN-10 | 0.540 ± 0.003 | 0.614 ± 0.002 | NN-10 | 0.081 ± 0.004 | 0.254 ± 0.002 |
| NNWang | 0.605 ± 0.002 | 0.657 ± 0.004 | NNWang | 0.029 ± 0.005 | 0.024 ± 0.000 |
| MMD | 0.618 ± 0.003 | 0.615 ± 0.008 | MMD | 0.151 ± 0.006 | 0.032 ± 0.001 |
| OSVM | 0.579 ± 0.001 | 0.538 ± 0.000 | OSVM | 0.249 ± 0.012 | 0.444 ± 0.009 |
| CNN | 0.901 ± 0.011 | 0.809 ± 0.029 | CNN | 0.699 ± 0.037 | 0.564 ± 0.060 |

threshold:

$$f(\mathbf{I}^*) = \min_{\mathbf{I}_i \in \mathcal{D}} \widehat{KL}(p_{\mathbf{I}^*} || p_{\mathbf{I}_i}).\tag{21}$$

**Defect localization** Once the defective surface is detected, the spot of the defect can be localized by inspecting the LPDR $\log(p_{\mathbf{I}^*}(\mathbf{x})/p_{\mathbf{I}_m}(\mathbf{x}))$ score between the query image $\mathbf{I}^*$ and the $\mathbf{I}_m$ with $\mathbf{I}_m = \arg\min_{\mathbf{I}_i \in \mathcal{D}} \widehat{KL}(p_{\mathbf{I}^*} || p_{\mathbf{I}_i})$. The location of the patch $\mathbf{x}$ with the large LPDR score is considered to be the defect location. Note that a similar approach has been used for the witness function in statistical model criticism [44, 45].

For the evaluation of the algorithm, we use a publicly available dataset for surface inspection: DAGM[2]. The dataset contains six distinct types of normal and defective surfaces. The defective samples are not used in training, but they are used in searching the decision thresholds. We extract 900 patches per image, and each patch is transformed into a four-dimensional feature vector. Then, the detection is performed and compared with many well-known criteria: diverse K-L divergences estimators as well as the maximum mean discrepancy (MMD) [46] and the one-class support vector machines (OSVM) [47]. In addition, the Convolutional Neural Networks (CNNs) training result is presented for comparison with a supervised method.

In DAGM, the testing data have defect patterns similar to those in the training data. To demonstrate unseen defect patterns, we artificially generate defective images by superimposing a randomly selected 15% of the normal testing images with a small image of the MNIST digit '3' at a random location (see Figure 6). Table 1 presents the area under curve (AUC) of the receiver operating characteristic curve for the detection as well as the mean average precision (mAP) for the localization.

CNNs which use labels for training show good performances only in detecting and localizing DAGM defects. The K-L divergence estimation with VWKDE show the best performance over many unsupervised methods, and it provides significantly better performances both at identifying unseen defects and at localizing them. Figure 6 shows one example of how well the proposed method localizes the position of the unseen defects.

## 6 Conclusion

In this paper, we have shown how a weighted kernel formulation for the plug-in densities could be optimized to mitigate the bias in consideration of the geometry of densities. The underlying mechanism uses the information from the first derivatives to alleviate the bias due to the second derivatives.

In our experiments, a simple choice of Gaussian density model for obtaining the first and second derivatives led to a reliable reduction of bias. This insensitivity to the exactness due to a coarse model is nonintuitive considering the traditional dilemma prevalent in many conventional methods; a coarse and inexact model enjoys a small variance but at the cost of large bias. In our work, the usage of the coarse model had no effect on the flexibility of the plug-in estimator, while the high dimensional bias was tackled precisely.

---

[2]Deutsche Arbeitsgemeinschaft für Mustererkennung (The German Association for Pattern Recognition).

Limitations of this study include the computational overhead for score learning using parametric or neural network methods and no benefit for the asymptotic convergence rate because it depends on the convergence rate of KDE. Using a non-flexible parametric model rather than a flexible one provides a consistent benefit to improve the KDE.

## Acknowledgments and Disclosure of Funding

SY was supported by a KIAS Individual Grant (AP095701) via the Center for AI and Natural Sciences at Korea Institute for Advanced Study. SY and FCP were supported in part by IITP-MSIT (2021-0-02068, 2022-0-00480), ATC+ (20008547), SRRC NRF (RS-2023-00208052), and SNU Institute for Engineering Research. GY was partly supported by IITP-MSIT (2019-0-01906), and IK and YKN was supported by NRF/MSIT (No. 2018R1A5A7059549, 2021M3E5D2A01019545), IITP/MSIT (IITP-2021-0-02068, 2020-0-01373, RS-2023-00220628). YKN was supported by Samsung Research Funding & Incubation Center for Future Technology (SRFC-IT1901-13) partly in the derivation of the weight dependency on the bias and its mechanical interpretation.

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

# Appendix

## A  Bias Derivation of the Posterior Estimator

The expectation of the weighted KDE is obtained from the following equation:

$$\mathbb{E}_{\mathcal{D}_1}[\widehat{p}_1(\mathbf{x})] \;=\; \mathbb{E}_{\mathbf{x}'\sim p_1(\mathbf{x})}[\alpha(\mathbf{x}')k_h(\mathbf{x},\mathbf{x}')] \;=\; \int \alpha(\mathbf{x}')p_1(\mathbf{x}')k_h(\mathbf{x},\mathbf{x}')\mathrm{d}\mathbf{x}' \tag{22}$$

$$= \int \alpha(\mathbf{x}')p_1(\mathbf{x}')\frac{1}{h^D}K\left(\frac{\mathbf{x}'-\mathbf{x}}{h}\right)\mathrm{d}\mathbf{x}' = \int \alpha(\mathbf{x}+h\mathbf{z})p_1(\mathbf{x}+h\mathbf{z})K(\mathbf{z})\mathrm{d}\mathbf{z}, \tag{23}$$

with the substitution $\mathbf{z}=\frac{\mathbf{x}'-\mathbf{x}}{h}$, or $\mathbf{x}'=h\mathbf{z}+\mathbf{x}$ to produce $\mathrm{d}\mathbf{x}'=h^D\mathrm{d}\mathbf{z}$, and $K\left(\frac{\mathbf{x}'-\mathbf{x}}{h}\right)=h^D k_h(\mathbf{x},\mathbf{x}')$ with normalized and isotropic $K(\mathbf{z})$. We apply Taylor expansion on the term $\alpha(\mathbf{x}+h\mathbf{z})p_1(\mathbf{x}+h\mathbf{z})$ around $\mathbf{x}$, and with the assumption that $|h\mathbf{z}|$ is small,

$$\alpha(\mathbf{x}+h\mathbf{z})p_1(\mathbf{x}+h\mathbf{z}) = \alpha(\mathbf{x})p_1(\mathbf{x}) + h\mathbf{z}^\top\nabla[\alpha(\mathbf{x})p_1(\mathbf{x})] + \frac{h^2}{2}\mathbf{z}^\top\nabla\nabla[\alpha(\mathbf{x})p_1(\mathbf{x})]\mathbf{z} + O(h^3), \tag{24}$$

using the Hessian operator $\nabla\nabla$. Now the integration yields the expectation with respect to $K(\mathbf{z})$ that satisfies $\int K(\mathbf{z})d\mathbf{z}=1$, $\int \mathbf{z}K(\mathbf{z})d\mathbf{z}=0$, and $\int \mathbf{z}\mathbf{z}^\top K(\mathbf{z})d\mathbf{z}=I$:

$$\mathbb{E}_{\mathcal{D}_1}[\widehat{p}_1(\mathbf{x})] = \alpha(\mathbf{x})p_1(\mathbf{x}) + \frac{h^2}{2}\nabla^2[\alpha(\mathbf{x})p_1(\mathbf{x})] + O(h^3), \tag{25}$$

with the Laplacian operator $\nabla^2$.

Along with the expansion for $\mathbb{E}_{\mathcal{D}_2}[\widehat{p}_2(\mathbf{x})]$, the following plug-in posterior can be perturbed by $h$ assuming a small $h$:

$$\mathbb{E}_{\mathcal{D}_1,\mathcal{D}_2}[f(\mathbf{x})] \;\rightarrow\; \frac{\mathbb{E}_{\mathcal{D}_1}[\widehat{p}_1(\mathbf{x})]}{\mathbb{E}_{\mathcal{D}_1}[\widehat{p}_1(\mathbf{x})] + \gamma\mathbb{E}_{\mathcal{D}_2}[\widehat{p}_2(\mathbf{x})]} \tag{26}$$

$$= \; f(\mathbf{x}) + \frac{h^2}{2}\frac{\gamma p_1(\mathbf{x})p_2(\mathbf{x})}{(p_1(\mathbf{x})+\gamma p_2(\mathbf{x}))^2}\left(\frac{\nabla^2[\alpha(\mathbf{x})p_1(\mathbf{x})]}{\alpha(\mathbf{x})p_1(\mathbf{x})} - \frac{\nabla^2[\alpha(\mathbf{x})p_2(\mathbf{x})]}{\alpha(\mathbf{x})p_2(\mathbf{x})}\right) + \mathcal{O}(h^3)$$

$$= \; f(\mathbf{x}) \; + \; \frac{h^2}{2}P(y=1|\mathbf{x})P(y=2|\mathbf{x})B_{\alpha;p_1,p_2}(\mathbf{x}) \; + \; \mathcal{O}(h^3), \tag{27}$$

giving the point-wise leading-order bias with respect to $h$:

$$\text{Bias}(\mathbf{x}) = \frac{h^2}{2}P(y=1|\mathbf{x})P(y=2|\mathbf{x})B_{\alpha;p_1,p_2}(\mathbf{x}). \tag{28}$$

Here, the $B_{\alpha;p_1,p_2}(\mathbf{x})$ is as follows:

$$B_{\alpha;p_1,p_2}(\mathbf{x}) \equiv \frac{\nabla^2[\alpha(\mathbf{x})p_1(\mathbf{x})]}{\alpha(\mathbf{x})p_1(\mathbf{x})} - \frac{\nabla^2[\alpha(\mathbf{x})p_2(\mathbf{x})]}{\alpha(\mathbf{x})p_2(\mathbf{x})}, \tag{29}$$

which includes the second derivative of $\alpha(\mathbf{x})p_1(\mathbf{x})$ and $\alpha(\mathbf{x})p_2(\mathbf{x})$. Because two classes use the same weight function $\alpha(\mathbf{x})$, Eq. (29) can be decomposed into two terms without the second derivative of $\alpha(\mathbf{x})$.

$$B_{\alpha;p_1,p_2}(\mathbf{x}) = \frac{\nabla^\top\alpha|_{\mathbf{x}}}{\alpha(\mathbf{x})}\left(\frac{\nabla p_1|_{\mathbf{x}}}{p_1(\mathbf{x})} - \frac{\nabla p_2|_{\mathbf{x}}}{p_2(\mathbf{x})}\right) + \frac{1}{2}\left(\frac{\nabla^2 p_1|_{\mathbf{x}}}{p_1(\mathbf{x})} - \frac{\nabla^2 p_2|_{\mathbf{x}}}{p_2(\mathbf{x})}\right). \tag{30}$$

## B  Solution of the Calculus of Variation

For the optimization of Eq. (15) with respect to $\alpha(\mathbf{x})$, we first make a substitution $\beta = \log\alpha$ and apply a calculus of variation technique for optimal $\beta(\mathbf{x})$. We express the objective functional with $\int m(\mathbf{x};\beta,\nabla\beta)\,d\mathbf{x}$ using

$$m(\mathbf{x};\beta,\nabla\beta) = \left(\nabla^\top\beta|_{\mathbf{x}}\mathbf{h}(\mathbf{x}) + g(\mathbf{x})\right)^2 r(\mathbf{x}). \tag{31}$$

With the substitution $\vec{\beta}' = \nabla\beta$ for notational abbreviation, we apply the Euler-Lagrange equation for the $m(\mathbf{x};\beta,\vec{\beta}')$ containing both $\beta$ and $\vec{\beta}'$:

$$\frac{\partial m(\mathbf{x};\beta,\vec{\beta}')}{\partial\beta} - \nabla_{\mathbf{x}}\cdot\nabla_{\vec{\beta}'}\,m(\mathbf{x};\beta,\vec{\beta}') = 0, \tag{32}$$

where the divergence is $\nabla_{\mathbf{x}} \cdot \nabla_{\vec{\beta}'} = \sum_{i=1}^{D} \frac{\partial}{\partial x_i} \frac{\partial}{\partial \beta'_i}$ with the $i$-th component of $\vec{\beta}'$, $\beta'_i$, and the dimensionality is $D$.

The first term can be calculated as $\frac{\partial m(\mathbf{x};\beta,\vec{\beta}')}{\partial \beta} = 0$. The first derivative of the second term is $\nabla_{\vec{\beta}'} m(\mathbf{x};\beta,\vec{\beta}') = r\left(\vec{\beta}'^{\top}\mathbf{h} + g\right)\mathbf{h}$. After we substitute $\beta(\mathbf{x})$ with $\log \alpha(\mathbf{x})$ back, we obtain the equation for the optimal $\alpha(\mathbf{x})$ function:

$$\nabla \cdot \left[ r((\nabla \log \alpha)^{\top}\mathbf{h} + g)\mathbf{h} \right] = 0. \tag{33}$$

which is Eq. (16).

## C   Analytic Solution for Two Homoscedastic Gaussians

We consider the following two homoscedastic Gaussians

$$p_1(\mathbf{x}) = \mathcal{N}(\mathbf{x}; \mu_1, \Sigma), \quad p_2(\mathbf{x}) = \mathcal{N}(\mathbf{x}; \mu_2, \Sigma), \tag{34}$$

with a common covariance matrix $\Sigma$.

In order to obtain the zero divergence in Eq. (16), the divergence-free vector field can be obtained using

$$\nabla \log \alpha \equiv \Sigma^{-1}\mathbf{x} + \vec{v}(\mathbf{x}) \tag{35}$$

because the inner product of $\nabla \log \alpha$ with $\mathbf{h} = \nabla \log p_1 - \nabla \log p_2 = \Sigma^{-1}(\mu_1 - \mu_2)$ should yield a negative value of $g(\mathbf{x})$, which is $-g(\mathbf{x}) = -\mathbf{x}^{\top}\Sigma^{-2}(\mu_2 - \mu_1) - \frac{1}{2}(\mu_1^{\top}\Sigma^{-2}\mu_1 - \mu_2^{\top}\Sigma^{-2}\mu_2)$. The equation $(\nabla \log \alpha)^{\top}\mathbf{h} = -g(\mathbf{x})$ gives

$$\vec{v}(\mathbf{x}) = -\frac{1}{2}\Sigma^{-1}(\mu_1 + \mu_2). \tag{36}$$

Therefore, one possible solution for $\nabla \log \alpha(\mathbf{x})$ is

$$\nabla \log \alpha(\mathbf{x}) = \Sigma^{-1}(\mathbf{x} - \mu), \tag{37}$$

with the mean of the two class-conditional means, $\mu = \frac{\mu_1 + \mu_2}{2}$. Therefore, one particular solution for $\alpha(\mathbf{x})$ is

$$\alpha(\mathbf{x}) = \exp\left[ \frac{1}{2}(\mathbf{x} - \mu)^{\top}\Sigma^{-1}(\mathbf{x} - \mu) \right]. \tag{38}$$

This solution is not unique, and any $\log \alpha$ that has a form of

$$\log \alpha(\mathbf{x}) = \frac{1}{2}(\mathbf{x} - \mu)^{\top}\Sigma^{-1}(\mathbf{x} - \mu) + l(\mathbf{x}), \tag{39}$$

with $l(\mathbf{x})$ satisfying $\nabla^{\top}l\,\Sigma^{-1}(\mu_1 - \mu_2) = 0$ is also the solution. One technique for finding such $l(\mathbf{x})$ is that we pick up any differentiable seed function $l_0(\mathbf{x})$ and consider its derivative $\nabla l_0$ with the $\vec{a} = \Sigma^{-1}(\mu_1 - \mu_2)$ component subtracted: $\left(I - \frac{\vec{a}\vec{a}^{\top}}{||\vec{a}||^2}\right)\nabla l_0$. The $l(\mathbf{x})$ is a function that its derivative satisfies $\nabla l = \left(I - \frac{\vec{a}\vec{a}^{\top}}{||\vec{a}||^2}\right)\nabla l_0$.

For example, if we choose $l_0(\mathbf{x}) = \mathbf{x}$, then $l(\mathbf{x})$ is a function that satisfies $\nabla l = \left(I - \frac{\vec{a}\vec{a}^{\top}}{||\vec{a}||^2}\right)\nabla l_0 = \mathbf{1} - \frac{\vec{a}^{\top}\mathbf{1}}{||\vec{a}||^2}\vec{a}$, and we can get $l(\mathbf{x}) = \left(\mathbf{1} - \frac{\vec{a}^{\top}\mathbf{1}}{||\vec{a}||^2}\right)^{\top}\mathbf{x}$. If we choose $l_0(\mathbf{x}) = \frac{1}{2}||\mathbf{x}||^2$, we get $l(\mathbf{x}) = \frac{1}{2}\mathbf{x}^{\top}\left(I - \frac{\vec{a}\vec{a}^{\top}}{||\vec{a}||^2}\right)\mathbf{x}$ after similar calculations. Now the choice of

$$l_0(\mathbf{x}) = -\frac{b}{2}(\mathbf{x} - \mu)^2, \tag{40}$$

gives us $l(\mathbf{x}) = -\frac{b}{2}(\mathbf{x} - \mu)^{\top}\left(I - \frac{\vec{a}\vec{a}^{\top}}{||\vec{a}||^2}\right)(\mathbf{x} - \mu)$, which produces our analytic weight function in Eq. (18):

$$\alpha(\mathbf{x}) = \exp\left( -\frac{1}{2}(\mathbf{x} - \mu')^{\top}A(\mathbf{x} - \mu') \right), \tag{41}$$

with $\mu' = \frac{\mu_1 + \mu_2}{2}$ and $A = b\left(I - \frac{\Sigma^{-1}(\mu_1 - \mu_2)(\mu_1 - \mu_2)^{\top}\Sigma^{-1}}{||\Sigma^{-1}(\mu_1 - \mu_2)||^2}\right) - \Sigma^{-1}$, with an arbitrary constant $b$.

## D   Posterior Prediction for Various Algorithms

Fig. 7 shows the posterior prediction results of various algorithms. For the estimation with support vector machines, [48] is used. For neural network estimation, 2-layer fully connected networks with 100 nodes for each layer were used minimizing the mean square error of the sigmoid output. Both VWKDE-MB and VWKDE-MF show superior results to other methods.

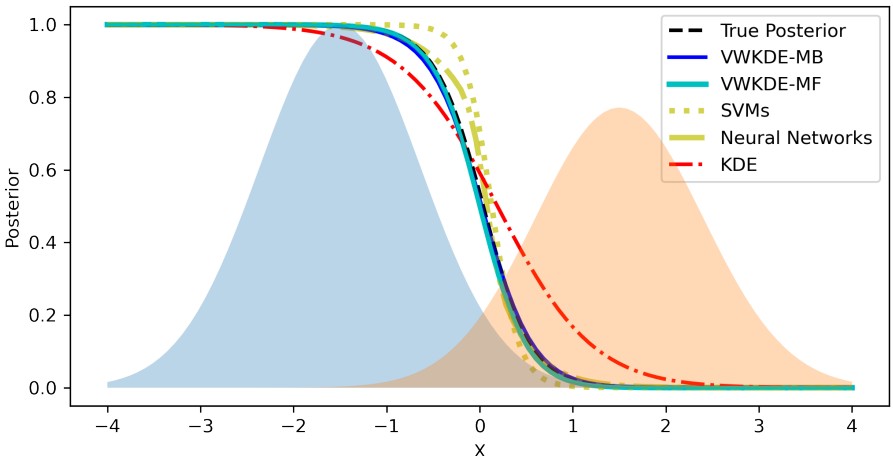

Figure 7: Posterior predictions with various algorithms for 20-dimensional Gaussians.

### D.1 Kernel density estimation in high dimensions

We also note the difficulty of density estimation with KDE in high dimensional space. Fig. 8 shows a one-dimensional slice of two 20-dimensional Guassians. The maximum density in this slice is on the order of $10^{-8}$. Meanwhile, the KDE with 5,000 data points per class shows densities on the order of about $10^{-10}$ with a bandwidth of $h = 0.8$.

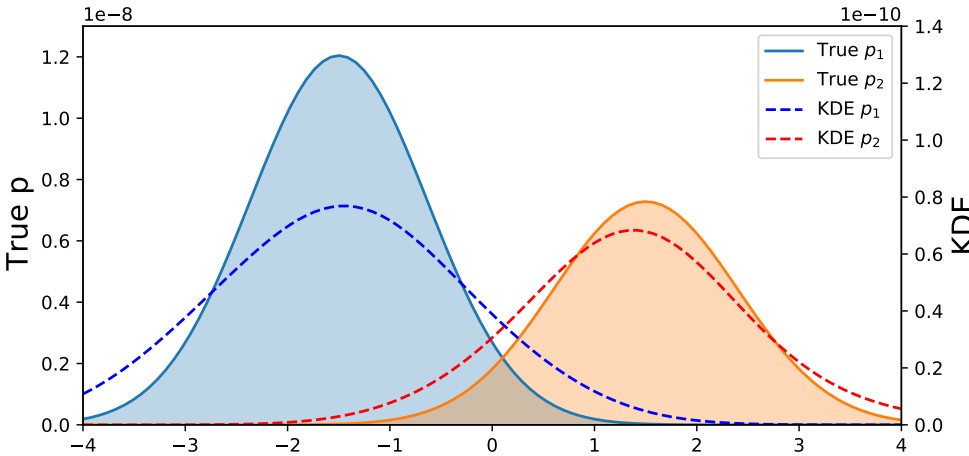

Figure 8: Underlying density functions and their KDE predictions.

The bandwidth should be chosen to be sufficiently large because no pairs are nearby in a high-dimensional space. Although the estimated density with $h = 0.8$ is reasonably smooth, the KDE differs from the true density by several orders of magnitude with 5,000 data points. However, the patterns of relative overestimation and underestimation depicted in Fig. 1 are evident, and the proposed method can be applied even with inexact KDEs.

## E  Fluid Flow Interpretation of Making Bias

The change of concentration $u(t)$ at time $t$ due to the convection and diffusion can be written as

$$\frac{\partial u}{\partial t} = -\mathbf{v}^\top \nabla u + D' \nabla^2 u, \tag{42}$$

with the direction of convection $\mathbf{v}$ and the diffusion constant $D'$. The first term represents the convection, and the second term represents the diffusion.

The bias Eq. (30) can be reformulated as

$$
\text{Eq. (30)} \quad = \quad \frac{\nabla^\top \alpha}{\alpha}\left(\frac{\nabla p_1}{p_1} - \frac{\nabla p_2}{p_2}\right) + \frac{1}{2}\left(\frac{\nabla^2 p_1}{p_1} - \frac{\nabla^2 p_2}{p_2}\right). \tag{43}
$$

$$
= \quad \left[\nabla^\top\left(\log\alpha + \frac{1}{2}\log p_1\right)\nabla\log p_1 + \frac{1}{2}\nabla^2\log p_1\right] \tag{44}
$$

$$
- \left[\nabla^\top\left(\log\alpha + \frac{1}{2}\log p_2\right)\nabla\log p_2 + \frac{1}{2}\nabla^2\log p_2\right]. \tag{45}
$$

$$
= \quad \frac{\partial u_1}{\partial t} - \frac{\partial u_2}{\partial t} \tag{46}
$$

Eq. (44) and Eq. (45) can be understood as the two flows with concentrations $u_1 = \log p_1$ and $u_2 = \log p_2$, respectively, and the convection direction for $u_1$ is $\mathbf{v}_1 = \nabla(\log\alpha + \frac{1}{2}\log p_1)$, and the direction for $u_2$ is $\mathbf{v}_2 = \nabla(\log\alpha + \frac{1}{2}\log p_2)$. Without the weight, the convection directions are $\mathbf{v}_1 = \frac{1}{2}\nabla\log p_1$ and $\mathbf{v}_2 = \frac{1}{2}\nabla\log p_2$ but with weight, they change to $\mathbf{v}_1 = \nabla(\log\alpha + \frac{1}{2}\log p_1)$ and $\mathbf{v}_2 = \nabla(\log\alpha + \frac{1}{2}\log p_2)$.

The role of $\alpha$ is to modify the directions of convection toward $\nabla\log\alpha$ together, and in the example shown in Fig. 1, the change of $u_1$ and $u_2$ due to diffusion are negative and positive, respectively. The direction of convection can control the change of $u_1$ and $u_2$, and the $\alpha$ makes the difference between $\frac{\partial u_1}{\partial t}$ and $\frac{\partial u_2}{\partial t}$ as small as possible.

## F    Prototype Classification Interpretation of the Weighted Kernel Methods in Reproducing Kernel Hilbert Space (RKHS)

The kernel algorithm for classification can be viewed as prototype algorithms in the RKHS due to the decomposition of positive definite kernel functions [31, 30, 29, Section 1.2]:

$$
k(\mathbf{x}, \mathbf{x}') = \langle\phi(\mathbf{x}), \phi(\mathbf{x}')\rangle, \qquad \phi(\mathbf{x}), \phi(\mathbf{x}') \in \text{RKHS}, \tag{47}
$$

with the inner product operator $\langle ., .\rangle$ defined in RKHS.

Given a dataset $\{\mathbf{x}_i, y_i\}_{i=1}^N$, $\mathbf{x}_i \in \mathbb{R}^D$, $y_i \in \{0, 1\}$, the classification using two prototypes $\mathbf{w}_1 = \frac{1}{N_1}\sum_{\{i;y_i=1\}}\alpha_i\phi(\mathbf{x}_i)$ and $\mathbf{w}_0 = \frac{1}{N_0}\sum_{\{i;y_i=0\}}\alpha_i\phi(\mathbf{x}_i)$ in RKHS determines which of the prototypes has a smaller distance to the $\phi(\mathbf{x})$ than the other. Here, $N_1$, $N_0$ are the numbers of data of classes 1 and 0, respectively. The classification using KDEs with the pointwise weights can be compared with the prototype classification in RKHS using the following derivation:

$$
y = \mathbf{1}\left(\left[\frac{1}{N_1}\sum_{\{i;y_i=1\}}\alpha_i k(\mathbf{x}_i, \mathbf{x}) - \frac{1}{N_0}\sum_{\{i;y_i=0\}}\alpha_i k(\mathbf{x}_i, \mathbf{x})\right] > \theta\right) \tag{48}
$$

$$
= \mathbf{1}\left(\left[\frac{1}{N_1}\sum_{\{i;y_i=1\}}\alpha_i\langle\phi(\mathbf{x}_i), \phi(\mathbf{x})\rangle - \frac{1}{N_0}\sum_{\{i;y_i=0\}}\alpha_i\langle\phi(\mathbf{x}_i), \phi(\mathbf{x})\rangle\right] > \theta\right) \tag{49}
$$

$$
= \mathbf{1}\left(\left[\left\langle\frac{1}{N_1}\sum_{\{i;y_i=1\}}\alpha_i\phi(\mathbf{x}_i), \ \phi(\mathbf{x})\right\rangle - \left\langle\frac{1}{N_0}\sum_{\{i;y_i=0\}}\alpha_i\phi(\mathbf{x}_i), \ \phi(\mathbf{x})\right\rangle\right] > \theta\right) \tag{50}
$$

$$
= \mathbf{1}\left(\left[\langle\mathbf{w}_1, \phi(\mathbf{x})\rangle - \langle\mathbf{w}_0, \phi(\mathbf{x})\rangle\right] > \theta\right) \tag{51}
$$

$$
= \mathbf{1}\left(\left[||\mathbf{w}_1 - \phi(\mathbf{x})||^2 - ||\mathbf{w}_0, \phi(\mathbf{x})||^2\right] > \theta'\right) \tag{52}
$$

with a predetermined threshold $\theta$. In Eq. (52), $\theta' = \theta - (||\mathbf{w}_1||^2 - ||\mathbf{w}_0||^2)$.

With uniform weight $\alpha_i = 1$ for all $i = 1, \ldots, N$, the classification is simply the comparison of two KDEs $\widehat{p}_1 = \frac{1}{N_1}\sum_{\{i;y_i=1\}}k(\mathbf{x}_i, \mathbf{x})$ and $\widehat{p}_0 = \frac{1}{N_0}\sum_{\{i;y_i=0\}}k(\mathbf{x}_i, \mathbf{x})$, which correspond to the prototype classification using two empirical means $\mathbf{w}_1 = \frac{1}{N_1}\sum_{\{i;y_i=1\}}\phi(\mathbf{x}_i)$ and $\mathbf{w}_0 = \frac{1}{N_0}\sum_{\{i;y_i=0\}}\phi(\mathbf{x}_i)$ in RKHS. Originating from this correspondence, one suggestion of the unification for the KDE and RKHS is presented in [30]. The explanations about the prototypes for various kernelized algorithms can be found in [49]. The prototypes of SVMs are known to be the closest two points within the convex hull of different classes [50].

Despite all these discussions, it is clear that the modification of the densities using a weight function will not improve the density estimation performance from the perspective of KDE. Despite the poor density estimation

performance, the modification improves the classification or information-theoretic measure estimation for the KDE plug-in algorithms, but not necessarily the KDE itself. The improvement is partly supported by the prototype models in RKHS.

# G   Least Square Approach for Binary Classification

Reducing the bias of the posterior equation in Eq. (3) corresponds to the least square of the prediction error. The optimal square error is achieved with the Bayes classifier, which classifies a datum according to the posterior probability. The posterior probability of $\mathbf{x}_0$ being generated from $p_1(\mathbf{x})$ can be written as:

$$P(y = 1|\mathbf{x}_0) = \frac{p_1(\mathbf{x}_0)p(y = 1)}{p_0(\mathbf{x}_0)p(y = 0) + p_1(\mathbf{x}_0)p(y = 1)} \tag{53}$$

$$= \frac{p_1(\mathbf{x}_0)}{\gamma p_0(\mathbf{x}_0) + p_1(\mathbf{x}_0)}, \tag{54}$$

where $\gamma = p(y = 0)/p(y = 1)$. The least square error with

$$L = \int (f(\mathbf{x}) - y)^2 p(\mathbf{x}, y) dy d\mathbf{x}, \tag{55}$$

is achieved with the following prediction function

$$f(\mathbf{x}) = \mathbb{E}[y = 1|\mathbf{x}] = P(y = 1|\mathbf{x}). \tag{56}$$

An accurate estimation of posterior is essential for successful classification. We construct a classifier based on the KDE density estimates $\widehat{p}_0(\mathbf{x})$, $\widehat{p}_1(\mathbf{x})$.

$$f(\mathbf{x}) = \frac{\widehat{p}_1(\mathbf{x})}{\gamma\widehat{p}_0(\mathbf{x}) + \widehat{p}_1(\mathbf{x})} \tag{57}$$

$$= \frac{1}{1 + \gamma(\widehat{p}_0(\mathbf{x})/\widehat{p}_1(\mathbf{x}))}, \tag{58}$$

and consider the deviation of $f(\mathbf{x})$ from the true $\mathbb{E}[y = 1|\mathbf{x}]$.

# H   Details on Optical Surface Inspection Experiments

We use a widely used public surface inspection dataset provided by DAGM[3] for experiments. The dataset contains six distinct textile surface types and associated defect types. There are 1,150 images per class, half of which is for training and the remaining is for testing. Approximately 13% of total images are defective, and for each defective image, a masking image which roughly encloses the defective region are provided. The dataset is originally proposed for a supervised setting.

We extract 900 patches of size 32×32 from each image, using a sliding window with step size 16. In each patch, we apply Gaussian smoothing and Scharr kernel to obtain a gradient distribution which can capture the texture information. To encode the gradient distribution as a feature vector, we compute its mean, standard deviation, skewness, and kurtosis. As a result, a surface image is transformed into a set of 900 four-dimensional vectors, or a 900×4 matrix. Feature vectors are standardized and whitened by aggregating all the patches from the same surface type.

VWKDE can be time-consuming as a large number of KL divergences need to be computed. Therefore, we take a two-pass approach when applying VWKDE. Given a query image, we first apply KDE-based KL divergence estimator to obtain rough estimates of KL divergences. Then, we take $k$ images with the lowest KL divergences and apply VWKDE-based KL divergence estimator to the $k$ images to finally select the image with the lowest KL divergence. This method enables us to have the best of both worlds, the speed of KDE and the accuracy of VWKDE.

The optimal bandwidth for bias reduction methods such as Ensemble [36], vonMises [37], and VWKDE is usually larger than other methods. We use the bandwidth with maximum leave-one-out log-likelihood of KDE for other methods but in these three methods, we used the heuristic rule of bandwidth selection using the maximum log-likelihood bandwidth for only 25% of randomly selected data.

A convolutional neural network (CNN) which takes a 32×32 patch as an input and predicts whether the patch is defective is trained. For training, we label patches with 75% overlap to the defect mask as defective and patches

---

[3]Deutsche Arbeitsgemeinschaft für Mustererkennung (The German Association for Pattern Recognition). Data access: https://hci.iwr.uni-heidelberg.de/node/3616

without any overlap to the defect mask as normal. Patches do not belong to either class are discarded. Due to class imbalance, normal patches are undersamples to yield defect to normal ratio of 1:4. CNN is trained for each surface type separately. The structure of our CNN is Conv(20)-Conv(20)-MaxPool-Conv(20)-Conv(20)-MaxPool-FC(20)-DropOut-FC(1), where Conv is a $3\times3$ convolution layer, MaxPool is a $2\times$ max pooling layer, FC is a fully connected layer, and DropOut is a drop out operation with probability 0.5. We use binary cross entropy loss for objective function and ADAM for optimization.

In unsupervised defect localization, we threshold log probability density ratio (LPDR) estimate to obtain detection results. We threshold LPDR estimates dynamically at 90% of maximum LPDR observed in the image. Then, we use its KL divergence estimate as a confidence score for the detection. For a CNN, we use the output probability for a patch as a detection score, and set a threshold to 0.9, and the maximum probability of defect among the patches in an image is used as a confidence score. Note that, in this experiment, we generate one detection per an image because DAGM dataset is constrained to have as most one defect per an image. However, this condition can be relaxed in future work with other dataset.

Intersection-over-union (IOU) is computed between a detection and a true defect mask. A detection with IOU larger than 0.1 considered as a correct detection. This threshold is lower than a typical threshold in object detection (0.5), because the defect mask is weakly labelled and usually larger than a precise defect region. Using a confidence score assigned for a detection, we compute average precision as in PASCAL VOC challenge, then take average over surface types.

