# OpenReview forum: "Variational Weighting for Kernel Density Ratios"
_NeurIPS.cc/2023/Conference — NeurIPS 2023 poster_

### Official Review · Reviewer_oTo7 · 2023-07-04

**Soundness:** 4 excellent
**Presentation:** 3 good
**Contribution:** 3 good
**Rating:** 7
**Confidence:** 3

**Summary:**

The paper presents an optimized weight function for kernel density estimations, which reduces bias and improves estimates of prediction posteriors and information-theoretic measures. The weight function is derived using tools from multidimensional calculus of variations and utilizes information from the first derivatives to mitigate bias caused by the second derivatives.

Experiments demonstrate the reliable bias reduction achieved by using a simple choice of a Gaussian density model for the derivatives. Interestingly, this bias reduction is independent of the accuracy of the model, addressing a common dilemma in conventional methods. Typically, using a coarse and inaccurate model results in low bias but high variance. In this case, the use of a coarse model does not affect the flexibility of the estimation procedure, while effectively addressing high-dimensional bias.

However, there are some limitations to consider, such as the computational overhead of score learning using parametric or neural network methods, and the lack of impact on the asymptotic convergence rate, which depends on the convergence rate of the kernel density estimation. Nevertheless, the use of a non-flexible parametric model consistently improves the kernel density estimation.


**Strengths:**

Content: The paper presents a novel approach to kernel density estimation using an optimized weight function. This method reduces bias and variance, addressing the traditional trade-off dilemma between bias and variance. The proposed method is independent of the model's accuracy and effectively handles dimension-dependent bias. Importantly, it does not compromise the flexibility of the estimation procedure. Overall, the paper stands out for its innovative approach, solid theoretical foundation, effective bias reduction, flexibility, and superior performance.

Formalism: The paper is well-written (up to minor details, see below), full of details, perhaps slightly too formalistic. It makes a non-trivial contribution seem easy to follow.

Experiments: Several experiments in various complex ways are being presented.


**Weaknesses:**

I have no major problems with the paper. While regularly using KDEs and all tools employed in the paper, I am not sure that I know all of the recent literature.

There are some minor things, that can easily be fixed for the camera ready version:
 - l. 23-27: Please mention here the cause of underestimation and overestimation in convex and concave regions (Why does it happen? Perhaps refer to Figure 1?).
 - Figure 1: In this figure, only the difference at a single point x_0 is depicted. Would it be possible to include the complete representation of $\hat{p_1}$ and $\hat{p_2}$ (dotted lines)? Here or elsewhere?
 - l. 30-35 or 36: Perhaps mention in the introduction that $\alpha$ is used as the product in the kernel density estimator.
 - l. 72-73: Explain $f$ in Equation 3. The symbol was not mentioned before.
 - Fig. 2: Write out the abbreviations (VWKDE) when they are used for the first time. Perhaps even state the name much earlier, like in the introduction or in the abstract?
 - Inclusion of code would have been nice. (However, the method does not seem to be difficult to implemented and does not strongly depend on hyperparameter choices. So this is not a major drawback.)

It is typical that the supplementary material does not reduplicate the paper. ;-)


**Questions:**

- How does VWKDE react to very small bandwidth for multimodal datasets really many observations? I am interested in KDE applications with such data.

**Limitations:**

No limitations are being stated. I do not see any limitation myself.

---

> ### Author Rebuttal · Authors · 2023-08-09
>
>
> Thank you for taking the time to review our paper. We deeply value your feedback.
>
>
> > l. 23-27: Mention the cause of underestimation and overestimation in convex and concave regions
>
> First, we will include Larry Wasserman (2005) *All of nonparametric statistics*, Springer, in our revised version as a reference for providing a standard derivation of the KDE bias. In the proof of Theorem 6.28 in this book, the following equation for the KDE bias can be found:
> \begin{eqnarray}
> \mathbb{E}[K_{h_n}(x, X)]  - f(x) = \frac{1}{2}\sigma_K^2 h_n^2 f’’(x) + O(h_n^4),
> \end{eqnarray}
> where $f(x)$ is the true density and $f’’(x)$ is its second derivative.
>
> The first two figures in Fig. 1 is an illustration of this equation. For explanation, we additionally assume a new density $p_3$ that has an exactly equivalent value to $p_1$ at $\mathbf{x}_0$, or $p_3(\mathbf{x}_0) = p_1(\mathbf{x}_0)$, but is uniform in the neighborhood of $\mathbf{x}_0$. Then the number of samples from $p_1$ should be smaller than the number of samples from $p_3$ around $\mathbf{x}_0$ because the integration of the density is the probability. The KDE counts the effective number of the neighborhood samples and is unbiased only for the samples from a uniform density throughout the domain near $\mathbf{x}_0$. Therefore, the KDE should underestimate $p_1(\mathbf{x}_0)$ while the KDE for $p_3(\mathbf{x}_0)$ is unbiased. The overestimation of $p_2(\mathbf{x}_0)$ can be explained similarly. The association of the second derivative and the bias is inevitable due to the nonzero bandwidth in the kernel.
>
>
> > Figure 1: not only at a single point x_0, include the complete representation of $\widehat{p}_1$ and $\widehat{p}_2$.
>
> In addition to Fig.2(a) and Fig3(a), we will include a figure in the Appendix that shows the true p_1, p_2, the estimated $\widehat{p}_1$, $\widehat{p}_2$, and the true and the estimated ratios for all points along a one-dimension span.
>
>
> > l. 30-36: Mention that \alpha is used as the product in the kernel density estimator
>
> We will update the introduction to highlight that the alpha is the weight being multiplied to kernels.
>
>
> > l. 72-73: Explain f in Equation 3. The symbol was not mentioned before.
>
> We will mention f(x) and its meaning before use.
>
>
> > State the abbreviation VWKDE early.
>
> We will write out the abbreviation VWKDE at the title of Section 2 and the abstract.
>
>
> > Inclusion of code
>
> Thank you for your suggestion. We will make the implementation of VWKDE publicly available through an online repository.
>
>
>
> > How does VWKDE react to very small bandwidth for multimodal datasets really many observations? I am interested in KDE applications with such data.
>
> As in conventional KDE, the bandwidth of VWKDE can be small if a large number of training data are provided. In theory, VWKDE can allow a larger bandwidth without sacrificing the mean square error because the weight function yields a bias reduction, which otherwise should be achieved by using a small bandwidth.

---

> ### Comment · Reviewer_oTo7 · 2023-08-10
> **Opinion after the rebuttals**
>
> I have carefully read all reviews and all rebuttals.
>
> My questions have been adequately answered.
>
> While the reviewers had additionally criticism from mine, I think the rebuttals address these points adequately and I think it plausible that the camera ready version is free of these points.
>
> My overall opinion remains as is, I think the paper should be published.

---

### Official Review · Reviewer_rQ6e · 2023-07-05

**Soundness:** 2 fair
**Presentation:** 3 good
**Contribution:** 2 fair
**Rating:** 6
**Confidence:** 2

**Summary:**

The paper studies the weighted kernel density estimation (wKDE) and proposes a method to learn the weights.

**Strengths:**

1. The authors illustrate biases in the standard KDEs and propose a mitigation by introducing weights in the KDE.
2. An estimator for the weight function is provided based on minimizing a quadratic loss.
3. Improved results are shown for the proposed method on the polluted MNIST data compared to several other approaches.

**Weaknesses:**

1. It seems that the bias occurs only when the data x is not sampled from the true distribution p(x). In this setting, the proposed method could reduce the bias. For x sampled from p(x), the standard KDE is a kernel mean embedding which is an unbiased estimator of the density.
2. The weighted KDE is essentially equivalent to the standard KDE with a kernel specified by the integral (Eq. 2). Thus, the weight learning may not outperform the standard KDE with an appropriate kernel.
3. The learning objective (Eq. 15) is based on a quadratic function, there could be better suited losses for densities.

**Questions:**

Instead of introducing the weights, why not learning a new kernel?

---

> ### Author Rebuttal · Authors · 2023-08-09
>
>
> Thank you for taking the time to review our paper. Below, we rebut the claims that the reviewer provided. We hope the concerns of the reviewer to be resolved.
>
>
> > The bias occurs only when the data x is not sampled from the true distribution p(x).
>
> This is not true. KDE should produce asymptotic bias with “nonzero bandwidth” even when x is from the true density.
>
>
> > For x sampled from p(x), the standard KDE is a kernel mean embedding which is an unbiased estimator of the density.
>
> No! The KDE is a “biased” estimator of the density. The nonzero bias exists even with infinite samples due to the nonzero bandwidth. Are you mentioning that the KDE equation is also used as the estimator of kernel mean embedding (KME)? In this case, it is an unbiased estimator of KME, not the unbiased estimator of the density. Could you clarify? The unbiasedness of KME estimator is found in the Eq.(3.12) in arXiv:1605.09522.
>
>
> > The weighted KDE is essentially equivalent to the standard KDE with a kernel specified by the integral (Eq. 2). Thus, the weight learning may not outperform the standard KDE with an appropriate kernel.
>
> Regarding the asymptotic convergence rate, the reviewer's assertion is true. Thus, the standard convergence proof can be applied interchangeably to both the conventional and the weighted KDE. However the asymptotic biases to which they converge are different with nonzero bandwidth. According to our derivation, the weighted KDE converges with a bias in Eq. (12), and the standard KDE converges with a bias which equals the second term in Eq. (12). Therefore, the first term in Eq. (12) is the difference that can make the weighted KDE outperform.
>
>
> > Quadratic objective function in (Eq. 15)
>
> The proposed quadratic objective produces the optimal condition for $\alpha(\mathbf{x})$ shown in Eq. (16) after applying the calculus of variation. Because of the special form in Eq. (16) regarding r(x), we could make a practical algorithm without considering r(x). If we use other objective functions, we expect many other objectives will produce an equation where r(x) is entangled.
>
>
> > Instead of introducing the weights, why not learning a new kernel?
>
> In Eq. (12), the first term can be controlled by weight, and we tried to make the whole bias in Eq. (12) disappear by controlling the first term. Without weight, the first term is a constant (zero). However as the reviewer suggested, learning the shape of a kernel that directly tackles the second term looks also valid. This motivation is similar to those tried in [25] though the research is performed in a different setting with nearest neighbor information. However, we expect the kernel learning to directly tackle the second term will produce similar results to those from [25] which we presented in Fig. 4.

---

> > ### Comment · Reviewer_rQ6e · 2023-08-18
> > **After rebuttal**
> >
> > Thank you for addressing my comments. I've increased my score accordingly.

---

### Official Review · Reviewer_QwPp · 2023-07-06

**Soundness:** 3 good
**Presentation:** 3 good
**Contribution:** 3 good
**Rating:** 6
**Confidence:** 2

**Summary:**

The authors propose an optimal weighting scheme to reduce bias in kernel density ratio estimation that alleviates the problem of under (or over) estimation due to the geometry of the distribution. They show that employing this weighting scheme leads to better estimates of prediction posteriors.

**Strengths:**

- The method is clearly motivated, and is theoretically sound (as far as I could tell).
- I am not an expert in the area and found the paper clear and easily accessible.
- I found the problem well motivated and the idea of adjusting the weight function variationally after estimating the first and second order properties of the underlying distributions to account for the under (or over) estimation interesting and well explored.
- The empirical analysis is detailed and shows the proposed approach leads to better estimation.

**Weaknesses:**

The proposed method is computationally more expensive. I am curious how the performance compares between different methods given the same amount of compute resources to make the comparison fair. Also, in the experimental results that are reported, what quantity has been kept constant across methods for the comparison? The weighting function does not improve convergence rates for KDE.


**Questions:**

See above.

**Limitations:**

Limitations have been discussed.

---

> ### Author Rebuttal · Authors · 2023-08-09
>
> Thank you for your constructive questions.
>
> > The proposed method is computationally expensive. Fair comparison
>
> The following is a table for the computation time with the 10D non-isotropic experiment in Fig. 4 for 4.2k samples per density:
>
> | Algorithm  | Time (s) |
> |------------|----------|
> | KDE         | 0.02       |
> | Ensemble |  1.68     |
> | Von          |   1.09    |
> | MINE       |   1.83    |
> | VWKDE_MB                                    |  2.22    |
> | VWKDE_MF Score model training  |  12.46  |
> | VWKDE_MF Weight model training |  73.7   |
> | VWKDE inference (with learned $\alpha(\mathbf{x})$) |  0.52   |
>
>
> The learning of weight function ($\alpha(\mathbf{x})$) is computationally expensive as pointed out by the reviewer. We will include the table of computation time in the Appendix for the readers who are interested in the fair comparison.
>
> From the perspective of practical usage, we note that VWKDE consists of two separable operations: the weight function learning and the nonparametric estimation with the learned weight. The weight function learning is a one-time operation, and the learned function can be reused. For instance, after training the $\alpha(\mathbf{x})$ function with a specific number of samples, we have the option to reuse the learned $\alpha(\mathbf{x})$ when we are capable of performing VWKDE with a larger dataset. While the results might not match those achieved with a retrained $\alpha(\mathbf{x})$, this approach offers a practical trade-off between time efficiency and increased accuracy. The VWKDE calculation time with a given $\alpha(\mathbf{x})$ is short as shown in the [VWKDE inference] in the above table.
> We will also include this discussion in the Appendix.
>
>
>
> > The weighting function does not improve convergence rates for KDE.
>
> The convergence rate has not improved. Only the asymptotic bias is improved.
> Our algorithm also provides many empirical factors on top of the derivation for bias reduction. We report that the proposed model-based and model-free methods empirically yield favorable outcomes when the direct information about the underlying densities and their derivatives are unavailable.

---

### Official Review · Reviewer_wV5E · 2023-07-12

**Soundness:** 2 fair
**Presentation:** 2 fair
**Contribution:** 3 good
**Rating:** 5
**Confidence:** 4

**Summary:**

The paper aims at improving the KDE estimator for the estimation of density ratios or KL divergences. To do so, they introduce a weighting function $\alpha$ whose role is to reduce the bias of the estimates. A variational problem is obtained using standard derivations, allowing to learn $\alpha$ in some model class (neural nets or RKHSs) using optimization principles.

The benefits of the approach are illustrated on synthetic and real benchmarks.

**Strengths:**

- The problem is of interest to the machine learning community
- The approach is novel
- The experiments section shows improvements over existing techniques

**Weaknesses:**

- The paper is not well written and lacks mathematical rigor (see questions)

**Questions:**

Here are a mix of questions/comments:

- [Major] The paper's structure could be greatly improved. In text derivations make the paper hard to read, and consume a lot of space for little gain. I would suggest to adopt an approach based on writing propositions highlighting the contributions, and making it easier to follow the logic behind the approach. Derivations without immediate interest could be put in the supplementary material. Moreover, what is the point of appendix H ? Either include it in the paper or drop it. Reorganizing the paper would free some space for these results.

- It would have been nice to have a reference for the paragraph at line 19-27 that explains the phenomenon (concave -> underestimation, convex -> overestimation).

- In general, please define symbols before you use them, e.g. for the plug-in posterior estimator there is no mention of labels or random variable $y$ before.
- Formally, $E_{D_1}$ makes little sense, $E_{x \sim p_1}$ would be better. The same holds for $E_{D_2}$ and $E_{D_1, D_2}$.
- The integration of $\mathcal{O}(h^3)$ at equation (7) is done without care. The local constants depend on $x$ so that there is an argument missing to just integrate.
- At equation (8), what does the right arrow mean ?
- At equation (9) $p_0$, $p_1$ should be $p_1, p_2$
- Argument for going from (8) to (9) is missing
- The assumptions regarding the kernel are not clearly stated (e.g. line 80). It would be good to have a \begin{assumptions} dedicated to the kernel from the beginning, so that the readers knows what kind of mathematical object is dealt with.
- At line 93: when you calculate the KL, why do you remove x_i ?
- At equation (15), it would be good to have some notion of the hypothesis space of $\alpha$
- In general, I find it weird to write an equality depending on $x$ on the left side and dropping it on the right side (e.g. equation 12).
- It is hard to understand equation (16) as it is. Is it for all $x, \alpha(x)$ should satisfy this ? Or is it from a functional point of view ? The answer can be found in the supplementary material, but the link is not made whith section B. Having a clearly stated proposition/proof in the appendix would solve this problem.
- At line 125: in general in the kernel literature, $\phi$ is the feature map and not the kernel, it would be good to respect this convention.
- I struggle to understand the difference between model free and model based ? To me there are two models: one is a NN, the other in a RKHS.
- Algorithm 1 could be cut in two parts, model free and model based, fitting the same vertical amount of space, allowing to detail the process for the model based optimization process.
- At line 135: one tries to minimize $B^2$ not $B$.
- Please use vectorial images allowing to zoom without blur, in general the plots could be greatly improved.
- Please use a standard font for the RKHS acronym.


**Limitations:**

Limitations have been addressed.

---

> ### Author Rebuttal · Authors · 2023-08-09
>
>
> We acknowledge the relevance of all the comments and have outlined our responses below, explaining how we intend to revise the manuscript in accordance with the feedback.
>
> > Major comment on the structure and the illustration of the main equations
>
> We will highlight the main equations and make sure their significant contributions come to the forefront. Part of the detailed derivation will be relocated to the Appendix, and we will let the sketch of the derivation techniques remain, so interested readers can derive the result themselves. We will get rid of Section H in the Appendix.
>
>
> > Reference for the phenomenon: concave -> underestimation and convex -> overestimation
>
> We will include chapter 6 of Larry Wasserman’s book (All of nonparametric statistics) as a reference. The standard derivation of the KDE bias includes the second derivative of the underlying density function. Though the association of the under- and overestimation and the concave and convex local density, respectively, is obvious from the literature and Fig. 1, the role of the weighting function to mitigate the bias proposed at line 28-35 stems from our original intuition and, to the best of our knowledge, have no prior reference.
>
>
> > Define symbols before you use them.
>
> We will check every symbol whether it is used before definition. It is correct that the variable y is used before definition.
>
>
> > Formally, E_{D_1} makes little sense, E_{x\sim p_1} would be better. The same holds for E_{D_2} and E_{D_1, D_2}.
>
> $D_1$ and $D_2$ are used to represent sets. For example in Eq. (4), the first and the second equation,
> $\mathbb{E}_{\mathcal{D}_1}[\cdot]$
>
> $=\mathbb{E}_{\mathbf{x}’\sim p_1(\mathbf{x})}[\cdot]$,
> is used to represent that the expectation in terms of the set $\mathcal{D}_1$ is equal to the expectation with respect to one variable $\mathbf{x}’$. Note that $\mathbf{x}$ is a fixed point, and $\mathbf{x}’$ is a random variable..
>
>
> > The integration of O(h^3) at equation (7) is done without care.
>
> We appreciate this correction! The comment is correct!! We will change O(h^4) to O(h^3) and change the writings accordingly including the definition of the kernels that we use.
>
>
> > The meaning of right arrow in Equation (8)
>
> Right arrow means the convergence. Asymptotically, each estimator in the numerator and denominator convergences.
>
>
> > At equation (9), p_0, p_1 should be p_1, p_2
>
> Thank you! The indexes are corrected.
>
>
> > Argument for going from (8) to (9) is missing
>
> The explanation will be included!
>
>
> > Assumption of the kernel
>
> This comment is completely relevant. We will include the mathematical assumption for the kernel.
>
>
> > At line 93: when you calculate the KL, why do you remove x_i ?
>
> The derived expectation $\mathbb{E}_{\mathbf{x}’\sim p}$ does not hold for $\mathbf{x}’ = \mathbf{x}_i$ because $\mathbf{x}_i$ is a fixed point for Monte-Carlo summation. We have to use $\mathbf{x}_i$ for $\mathbf{x} = \mathbf{x}_i$.
>
>
> > Hypothesis space for \alpha at equation (15)
>
> Thank you. $\alpha$ is a positive differentiable function. We will include the hypothesis space.
>
>
> > Argument x in the right hand side of equation (12)
>
> We will include x in every derivative and function on the right hand side.
>
>
> > Explanation for equation (16) and the link to Section B.
>
> The alpha function should satisfy Eq. (16) for all x. We will include this statement and write down a detailed derivation from Eq. (23) to Eq. (24) in Section B of the supplementary material and write down in the main text to refer to Section B.
>
> > Do not use \phi for kernel function and respect the convention
>
> Because $k$ is reserved for the kernel in KDE, we will use $\kappa$ to represent the RKHS basis.
>
>
> > Model-based and model-based: are they simply a function in RKHS and a NN?
>
> The name model-based does not come from the construction of a function in RKHS. For model-based function, we used a strong assumption that p_1 and p_2 are both Gaussians (L121). Model-free does not assume such models.
>
> > Algorithm 1
>
> We will make room to elaborate the model-based method in Algorithm 1.
>
>
> > B -> B^2
>
> Yes, corrected.
>
>
> > Vectorial image
>
> Figs 1-3 are not vectorized. We will make vectorial images for those figures.
>
>
> > RKHS acronym font
>
> We will use the standard font.

---

> > ### Comment · Reviewer_wV5E · 2023-08-21
> > **Acknowledging rebuttal**
> >
> > I thank the authors for their detailed answer. I believe that with the substantial changes agreed by the authors, the paper can make a decent addition to the conference.

---

### Decision · Program_Chairs · 2023-09-21

**Decision:**

Accept (poster)

**Comment:**

Reviewers agree about the novelty, structure, and the results of the paper.
Therefore the paper is recommended for acceptance with the changes discussed in the rebuttal.